# Kernel Regression with Infinite-Width Neural Networks on Millions of Examples

## Abstract

While kernel regression remains an important practical method, its connection to neural networks as their width becomes large has initiated fresh research. These neural kernels have drastically increased performance on diverse and nonstandard data modalities but require significantly more compute, which previously limited their application to smaller datasets. We address this by massively parallelizing their computation across many GPUs. We combine this with a distributed, preconditioned conjugate gradients algorithm to enable kernel regression at a large scale (i.e. up to 5 million examples). Using this approach, we study scaling laws of several neural kernels across many orders of magnitude for the CIFAR-5m dataset. Using data augmentation to expand the original CIFAR-10 training dataset by a factor of 20, we obtain a test accuracy of 91.2% (SotA for a pure kernel method). Finally, we explore other data modalities, obtaining results on protein and small molecule prediction tasks that are competitive with SotA methods.

## 1 Introduction

Kernel methods are often contrasted with deep learning, but recent advances in machine learning have identified and developed exciting correspondences (Lee et al., 2018; Matthews et al., 2018; Jacot et al., 2018). While a useful method in its own right, kernel regression has been used to better understand neural networks and deep learning. More specifically, if the parameters of a neural network are treated as random variables whose distribution is set by the initialization, we can view the neural network as a random function. Then as the width of the network becomes large, the distribution of this random function is a Gaussian process with a specific covariance function or kernel. We refer to kernels that arise from this connection with infinite-width neural networks as *neural kernels*. The specific kernel is determined by the architecture, inference type and other hyperparameters of the neural network.

Moreover, the connection between neural networks and Gaussian processes has generated many high-performance kernels for diverse or nonstandard data modalities, such as images, sequences, and graphs. This performance often comes at a cost, as the kernels require significantly more compute than standard kernels such as RBFs. For example, computing the entire kernel for the CIFAR-10 dataset takes less than 1 GPU minute for an RBF kernel but around 300 GPU hours for the Myrtle kernel (Shankar et al., 2020; Lee et al., 2020). However, this increase in compute significantly decreases the test error rate from around 40% to 10%. The added demands of simply computing entries of the kernel is in addition to challenges posed from the cubic scaling in time and quadratic scaling in memory of inference for kernel regression with dataset size. Approximate inference methods frequently reduce memory requirements by recomputing kernel entries on the fly, which is infeasible for these expensive kernels. Such challenges have limited our understanding of infinite-width models to small datasets. In particular, while scaling laws have been studied across many orders of magnitude for neural networks, the same is not true of their corresponding kernels. Similarly, while it is common to augment training datasets in neural networks, significantly expanding their size, the benefits for kernel methods have not been as thoroughly explored.

### 1.1 Contributions

In this work, we address these two main computational challenges (computing large, complex kernels and using them for kernel regression) by parallelizing and scaling up existing algorithms to

many more machines. This enables us to consider significantly larger datasets, currently up to five million examples, and therefore study how performance changes as more data are added over many orders of magnitude. While similar studies have been performed for neural networks, they have been lacking for neural kernels. In addition to scaling to larger datasets, we also consider high resolution images from the Tiny Imagenet dataset, where additional pixels also require more compute. Moreover, our approach is not restricted to image data. In fact, we obtain results for both protein sequence and small molecule datasets and demonstrate that neural kernels are a promising method for medium sized datasets from basic science.

Our contributions include:

- By massively parallelizing the computation of the neural kernels, we study kernels on significantly larger (approximately two orders of magnitude) datasets;
- We use a distributed, preconditioned conjugate gradients algorithm to perform inference for these kernels;
- We demonstrate scaling laws across several orders of magnitude for fully-connected, CNN-Vec, and Myrtle kernels on the CIFAR-5m dataset;
- We study the loss in performance incurred by approximating the linear system from inference compared to conjugate gradients;
- Using data augmentation to expand the original CIFAR-10 training dataset by a factor of 20, we obtain a test accuracy of 91.2% (SotA for a pure kernel method);
- We explore other data modalities, obtaining results on protein and small molecule prediction tasks that are competitive with SotA methods.

## 1.2 RELATED WORK

The contributions of this work are made possible by recent advances in large scale inference for Gaussian processes and kernel regression and advances in the understanding of the relationship between GPs and deep neural networks.

Wang et al. (2019) showed how to solve the large-scale linear systems within GPs using Conjugate Gradients (CG), which requires only matrix-vector operations with the kernel and doesn't require storing the full kernel in memory. Wang et al. (2019) and Maddox et al. (2022) identify the importance of using a pre-conditioner (a partially-pivoted Cholesky decomposition) to solve the system with finite precision. We use a similar CG algorithm to solve our systems and found that the preconditioner was critical to convergence, particularly for the non-stationary neural kernels. Due to their eigenspectra, we unfortunately found we required high precision on CPU. In this work, we use these methods to solve even larger systems, up to 5 million examples, but emphasize that the most computationally expensive component is computing the expressive neural kernels.

Many methods have been developed to approximate Gaussian processes on larger datasets. Rahimi & Recht (2007) show that stationary kernels can be approximated using a finite set of random basis functions. Recently there has been progress in random-feature approximations for expressive, non-stationary kernels using sketching (Zandieh et al., 2021; Han et al., 2022). While this method provides efficient ways for approximating neural kernels, often there is speed-performance trade-off (Han et al., 2022), thus our work focuses on exact computation. Stochastic variational inference (SVI) (Hensman et al., 2013) is a tremendously promising approach to scale GPs by optimizing a set of inducing points to approximate the full GP posterior. However, Wang et al. (2019) found that their SVI baseline underperformed exact GP inference in their experiments. We found that SVI was computationally infeasible due to taking gradients through the neural kernels, and using subsets of data as inducing points was less-effective than the full kernel. Other approaches, such as KISS-GP (Wilson & Nickisch, 2015; Stanton et al., 2021), cleverly interpolate over a grid of inducing points, taking advantage of Toeplitz structure, to drastically reduce computational cost.

Rudi et al. (2017) develop FALKON, which uses Nystrom and random feature approximations with pre-conditioning to scale up kernel regression to millions of examples with theoretical guarantees on performance. This is extended to potentially billions of examples by Meanti et al. (2020), through distributed GPU acceleration and various optimizations. EigenPro and EigenPro2 (Ma & Belkin, 2017; 2019) accelerate and scale up kernel regression by developing pre-conditioned stochastic gradient descent. While we observe that neural kernels scale remarkably well in terms of predictive

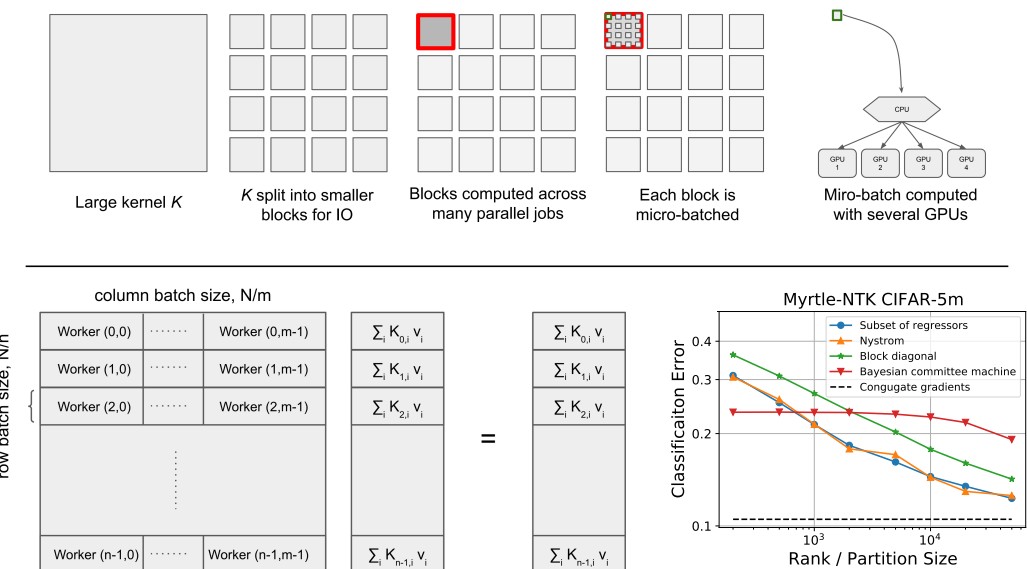

Figure 1: **(Top)** An upper diagonal of large kernel matrix is split into smaller, dimension 5,000×5,000 blocks for multi-threading of IO. When computing the kernel, the blocks can be computed independently by machines that need not be co-located. Each block is then batched further (to meet device memory limitations) and computed by the Neural Tangents library on several GPUs. **(Bottom left)** It is natural to split the kernel by rows across the workers, then each worker can receive the vector from the host, compute the matrix product between the rows in its memory with the vector, and return this chunk of the result to the host. The host can then aggregate the results from the workers by concatenating. However, for larger systems with many workers, simply communicating a large vector to all workers becomes a bottleneck. Instead the kernel can also be partitioned over columns. This means that each worker only needs a subset of the vector entries, and we found this dramatically increased communication speed. The cost of this approach is that slightly more complex aggregation is required on the host. **(Bottom right)** Classification error rate as a function of rank or partition size for four different kernel approximations. The kernel used is a 10-layer Myrtle NTK for 1.6 million examples of the CIFAR-5m dataset. The performance of CG is shown as a dashed line for comparison and exceeds all other results.

performance, using these methods could make them more computationally practical. However, we found that we have required the full kernel at high precision to obtain high performance.

The correspondence between random functions from the initialization of infinite-width neural networks and Gaussian processes, called the neural network Gaussian process (NNGP), was first noted by Neal (1994) and analytically derived by Williams (1996) for single-hidden layer, fully-connected networks. More recently this was extended to deep neural networks for fully-connected (Lee et al., 2018; Matthews et al., 2018) and convolutional architectures (Novak et al., 2019; Garriga-Alonso et al., 2019). This correspondence to Gaussian processes can be extended to infinite-width networks trained by gradient descent via the Neural Tangent Kernel (NTK) (Jacot et al., 2018) and to many other architectures, (Yang, 2019; 2020) including self-attention layers (Hron et al., 2020) and simple graph neural networks (Du et al., 2019). The covariance functions or kernels associated to these Gaussian processes are determined by the neural network's architecture and other hyperparameters from initialization, and we refer to them as neural kernels.

While these kernels are theoretically useful to better understand the functional prior of neural networks, including their uncertainty properties (Adlam et al., 2020), they have also been used more practically. The NTK for convolutional architectures has been applied to image data (Arora et al., 2019; Li et al., 2019; Shankar et al., 2020) and offers unique solutions to practical deep learning problems such as data distillation (Nguyen et al., 2021a;b). The JAX-based (Bradbury et al., 2018) python library, Neural Tangents (Novak et al., 2020), has helped to enable these practical applications by providing efficient computation of neural kernels on CPU, GPU and TPU.

## 2   APPROACHES TO LARGE-SCALE KERNEL METHODS

In this section, we review the different approaches to large-scale kernel regression. We discuss the particular challenges introduced by neural kernels and how we addressed them. Finally, we compare the performance for the CIFAR-5m dataset with 10-layer Myrtle NTK.

There are two main approaches to applying kernel methods to larger datasets, i.e. to solve a large linear system. First, the original linear system can be replaced by an alternative linear system that has a particular algebraic form that can be solved exactly and efficiently. Second, the original linear system can be maintained but only solved approximately.[1] These approximate solvers are often any-time algorithms, where the current solution can be returned at anytime—typically when its residual is sufficiently small.

**Approximations to the linear system:**   Many methods fall under this category. Here we focus on four different types. First, there are the low-rank approximations, *Nyström* and *Subset of Regressors*, that can be inverted in time $\mathcal{O}(r^2 n)$ where $r$ is the rank and $n$ is the number of training examples (Williams & Rasmussen, 2006).

Second, there is the *Block Diagonal* approximation that sets all kernel entries outside of some blocks along the diagonal to zero. Due to its algebraic form, this kernel is easy to invert and takes time $\mathcal{O}(r^2 n)$ where $r$ is the block or partition size. Even without parallelization, this reduces the running time by a factor of the number of blocks squared.

Finally, the *Bayesian Committee Machine* (Tresp, 2000; Williams & Rasmussen, 2006) is an approximation to the likelihood of a Gaussian process that partitions the dataset, fitting separate regressors to the partitions under assumptions of independence. Here we use the posterior mean under this approximation, which can be computed efficiently. Since this is a transductive method, the time complexity depends on the size of the test set but is comparable to the methods above when the number of test examples equals the partition size.

One advantage of these methods is that they do not require access to all kernel entries, and so can avoid the quadratic scaling from computing and storing the entire kernel. The primary additional hyperparameters in these methods are the rank or partition size. Typically increasing these hyperparameters improves performance at the cost of additional compute (see Fig. 1). However, in addition to their size, exactly which examples are included in the low-rank approximation or how the data are partitioned can affect performance. While several methods are available to select these examples more optimally, the most common approach is still to select uniformly at random, which we do here also. In addition, it was necessary to tune the jitter term in the *Nyström* approximation, since setting to the default value of $10^{-6}$ we used elsewhere yielded poor results.

**Iterative solvers:**   Rather than solving the linear system to machine precision, we might be satisfied with a solution that had a sufficiently small residual—especially because the predictions are unlikely to change significantly. Conjugate gradients (CG) is one approach to this (Shewchuk et al., 1994; Wang et al., 2019; Meanti et al., 2020). CG iteratively finds the best solution in a subspace of dimension equal to the number of steps. This subspace is defined by a sequence of vectors that are conjugate with respect to the kernel. Each step of CG takes quadratic time, and in the worst case a linear number of steps must be performed, but this is normally not necessary. To avoid this worst-case performance, pre-conditioning is essential. We used the standard partial pivoted Cholesky pre-conditioner (Wang et al., 2019) and found it significantly decreased the number of steps required to reach a small residual.

A key subroutine in CG is the computation of kernel-vector products and is the only place the kernel occurs directly. This presents opportunities for parallelization. In particular, we use a host CPU to run the main CG algorithm and act as a server to control worker CPUs who compute kernel-vector products (see Fig. 1 (Bottom left) for details).

**Additional challenges from neural kernels:**   Previous large-scale implementations of CG have traded off memory and time by never storing the whole kernel in memory and instead recomputing it on the fly. While this approach works well for simple kernels, such as RBFs, it is infeasible for neural kernels. In particular, computing the kernel once represents the majority of the compute com-

---

[1]Note that some approaches can be interpreted in both ways.

pared to running CG for inference. Indeed reducing the computational burden of neural kernels is a topic of current research and can be faster without decreasing performance (Zandieh et al., 2021; Han et al., 2021), though often there is a speed-performance tradeoff (Han et al., 2021). The problem has not been completely solved. Moreover, there is evidence that even computing them at `float32` precision compared to `float64` precision degrades performance (Lee et al., 2020). Thus the approach we take here to obtaining the kernel for larger datasets is to parallelize their computation across many machines (see Fig. 1 (Top) for details). Fortunately very little communication or coordination is needed across machines, and preemption-safe routines are easy to design. The kernel can then be computed once and stored on disk in a collection of sub-blocks for faster, multi-threaded IO. During CG, the kernel must be stored either completely in RAM, which is fastest but requires many machines for large kernels, or can be stored on disk with very fast IO to load chunks of the kernel into RAM.[2]

The spectra of neural kernels can be incredibly ill-conditioned (Lee et al., 2020), which presents challenges for inference and reinforces the need for pre-conditioning. Often the spectra are very diffuse, spanning several orders of magnitude and showing power-law decay.

**Performance comparison:**  In Fig. 1, we compare the accuracy of the different methods for the 10 layer Myrtle NTK on 1.6 million training examples from the CIFAR-5m dataset.[3] We find that as the rank or partition size increases, the performance of all methods improves with *Subset of Regressors* and *Nyström* performing best. We did not increase above 50,000 as this is close to the limit of the largest linear system that can be solved on a standard CPU. Note that while the gap to CG is reduced, there is still a large reduction in performance.

## 3  SCALING LAWS FOR NEURAL KERNELS

Known as *Neural Scaling Laws*, there has been substantial recent interest in how the performance of neural networks improves as more data are available for training and their number of parameters increases (Hestness et al., 2017; Kaplan et al., 2020; Rosenfeld et al., 2019; Rosenfeld, 2021; Bahri et al., 2021). Frequently, these quantities increase in tandem. Empirically, it has been found that performance improves as a power law, for which many works seek to estimate the exponent. Projecting the potential improvement in performance from scaling up neural networks is especially important in the large model era, where training such models is costly.

It is natural to ask whether other models also scale as a power law, and if so, how their exponents compare to those of neural networks. This comparison is particularly natural for neural kernels. Moreover, since they are nonparametric, their capacity scales automatically as more data are added. From the language of Bahri et al. (2021), dataset scaling for neural kernels is in the *resolution-limited* regime, which is arguably a more interesting scaling regime.

Scaling experiments for neural networks typically cover datasets over many orders of magnitude, which is challenging for kernel regression. In this section, we consider kernel regression for three different neural kernels: fully-connected neural networks, convolution neural networks without pooling operations, and the Myrtle convolutional networks. The kernels in that order increase in performance and computational burden, and all have been well tuned for the CIFAR-10 (Lee et al., 2020).

Of course, CIFAR-10 is limited in size with only 60k examples (50k train, 10k test). To enable a larger scaling study, instead we use the CIFAR-5m dataset (Nakkiran et al., 2021), which consists of samples from a generative model trained on CIFAR-10. Using CIFAR-5m, we can extend our analysis over 2 more orders of magnitude while ensuring the additional data are sampled i.i.d.. For generalization performance, in addition to evaluation on the i.i.d. CIFAR-5m held out data, we consider the CIFAR-10 and 10.1 test sets. Evaluation on these test sets is standard and allows for comparison against existing results.

In Fig. 2, we observe persistent dataset scaling laws as we vary architecture and evaluation data. For extremely small dataset sizes (10-100), all cases commonly show a lower slope and then transition

---

[2]For example, our 5 million examples kernel computed in `float64` precision, storage on disk alone is about 100 terabytes distributed over ∼500,000 blocks.

[3]See the next section for more details on this dataset.

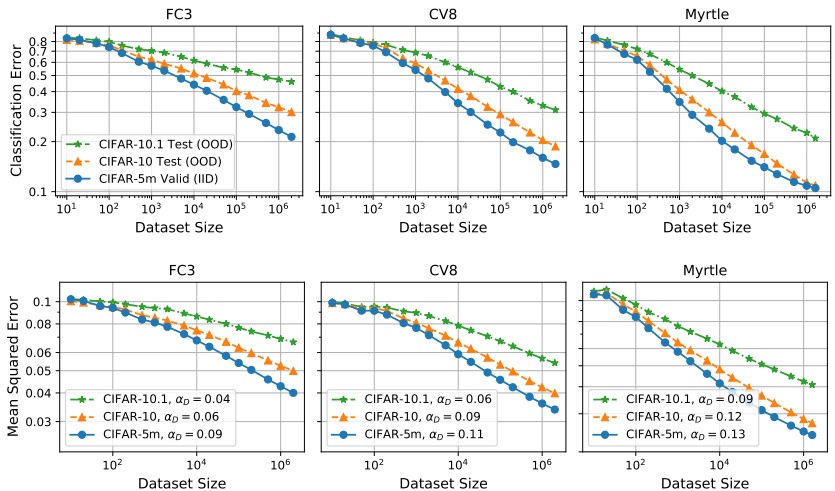

Figure 2: **Dataset size scaling for neural kernels on CIFAR-5m**. Using CIFAR-5m as training data, we explore dataset scaling spanning 6 orders of magnitude. Each column corresponds to different neural architectures: 3-layer fully-connected neural network (FC3), 8-layer convolutional neural network with output vectorization (CV8), and 10-layer Myrtle convolutional neural network with average pooling (Myrtle). Evaluations on three different held-out test sets, CIFAR-5m validation, CIFAR-10 test set, and CIFAR-10.1 test set, are shown. Scaling exponents on mean squared error are larger (thus faster improvement with more data) for more complex and computationally intensive kernels and for more in-distribution held out data (CIFAR-5m > CIFAR-10 > CIFAR-10.1).

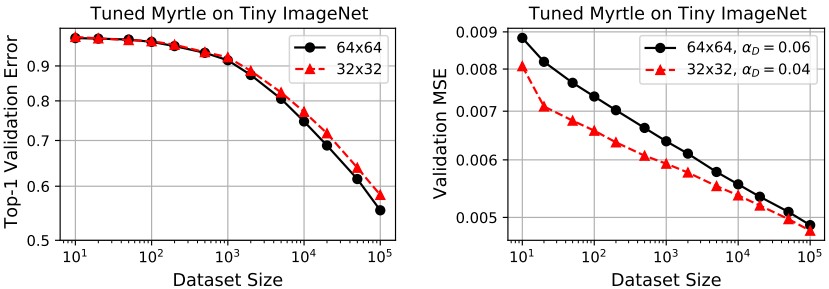

Figure 3: **Dataset scaling for Myrtle on Tiny ImageNet.** At full resolution (64×64), we achieve a best accuracy of 44.7%. Higher resolution inputs result in better classification accuracy and higher scaling exponent($\alpha_D$).

into a more consistent, higher slope, indicating power-law behavior across 4-5 orders of magnitude. We measure the dataset scaling exponent $\alpha_D$ for the test loss and observe that, as one increases the complexity of the kernel, the scaling exponent increases. Also, for the same architecture, evaluation data that are closer in distribution to the training data have a higher scaling exponent. Since the training data are drawn i.i.d from CIFAR-5m, the order of in-distribution to out-of-distribution for the test sets should be thought of as CIFAR-5m, CIFAR-10, and then CIFAR-10.1.[4]

**Tiny ImageNet: Towards ImageNet with neural kernels.** With our ability to compute massive kernels and use them for inference, ImageNet is not far from reach. ImageNet at 1.2 million examples is actually smaller than some datasets considered here—although augmentation beyond horizontal flips maybe challenging. However, at least two issues remain. First, the large number of output classes. Compared to the 10 classes in CIFAR-10 or CIFAR-5m, a one-hot label of 1k classes expands the linear systems to be solved by a factor of 100. Second, the resolution of Ima-

---

[4]This is because the generative model to generate CIFAR-5m was trained on the CIFAR-10 training set.

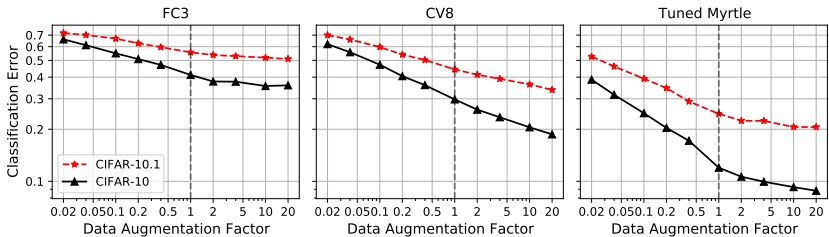

Figure 4: **Neural kernel performance on CIFAR-10 with data augmentation**. Using horizontal flip and RandAug, we measure scaling of neural kernels' classification error for 3 different architectures and 2 test sets (CIFAR-10 as well as CIFAR-10.1). Data augmentation factors smaller than 1.0 (light blue shaded region) denotes a subset of original training set, so additional i.i.d. data can be compared with augmented data. The augmentation factor 2× includes horizontal flips, whereas factors larger than 2× use RandAug combined with random crop and flips. While data augmentation consistently improves scaling, there is a noticeable change in slope around 1.0 for FC3 and Tuned Myrtle kernels (see Appendix C). For augmentation factors of 10× and 20×, the Tuned Myrtle kernel achieves less than 10% classification error (8.8% with 20×). (See Table 1 for detailed comparison.)

geNet in standard deep learning architecture is 224×224, if not larger (c.f. EfficientNet Tan & Le (2019; 2021)). For neural kernels with pooling, such as the Myrtle kernel, compute scales quadratically with the total number of pixels. Thus compared to 32×32 CIFAR-10 images, each data point requires $7^4 = 2401$ times more compute.

As a step toward ImageNet, we consider the Myrtle kernel for the Tiny ImageNet dataset (Le & Yang, 2015). Tiny ImageNet consists of 200 subclasses of ImageNet with 500 images per class, i.e. 100k total training images at a lower resolution of 64×64. We experimented with further downsizing to 32×32 resolution. Dataset scaling results are shown in Fig. 3. Similar to CIFAR-5m scaling, from a sufficiently large dataset size there is consistent power-law scaling. To the best of our knowledge, this is the first evaluation of neural kernels on Tiny Imagenet. We achieve at best classification accuracy of 44.7% on the test set. Comparing to finite networks, among very few references quoting results without data augmentation, Jeevan et al. (2022) obtained 43.1% for ResNet-18 and 45.39% for ConvMixer-16, indicating our result from neural kernel is competitive to modern finite neural network architectures without data augmentation.

## 4 DATA AUGMENTATION

Data augmentation is widely used in deep learning applied to image data. For example, for CIFAR-10 or ImageNet classification, most if not all models are trained using random crops and horizontal flips. More effective augmentation strategies such as AutoAug (Cubuk et al., 2019), RandAug (Cubuk et al., 2020) have been found and used in SotA vision models.

The idea of data augmentation has a long history (Niyogi et al., 1998)— including in SVMs where they are called virtual examples (Schölkopf et al., 1996). However, despite the broad application of data augmentation in deep learning, recently it has been used little in kernel methods due to their limitations on large datasets. For example, Li et al. (2019) and Shankar et al. (2020) used horizontal flips to double the training set size for convolutional kernels. In Lee et al. (2020), an ensemble of kernel predictors, where each training set was augmented randomly, were used as a means of data augmentation.[5] Because of this, exactly how much of the performance gap between neural networks and kernel methods can be accounted for by data augmentation on image data remains unknown.

In this section, we use our massively parallelized computation of neural kernels and distributed CG solver to explore the effect of data augmentation in neural kernels to a regime far beyond prior work. In our experiments, we focus on the CIFAR-10 dataset. We call the ratio between the sizes of the original training set and the training set after augmentation the *augmentation factor*. Our augmentation strategy is to expand the 50k training set by applying various augmentations. Following Li

---

[5]In effect, this is performing a block diagonal approximation on the augmented dataset.

| Architecture | | Method | CIFAR-10 | CIFAR-10.1 | CIFAR-5m |
|---|---|---|---|---|---|
| FC | Kernel | DA Ensemble (Lee et al., 2020) | 62.4 | - | - |
| | | DA CG (20x, this work) | 64.1 | 49.1 | - |
| | | CIFAR-5m CG (2M, this work) | **69.9** | **54.1** | 78.6 |
| | Finite NN | DA, Small LR, no L2 (Lee et al., 2020) | 65.3 | - | - |
| | | DA, Large LR, L2 (Lee et al., 2020) | 69.4 | | |
| CNN-VEC | Kernel | Flip (Li et al., 2019) | 70.5 | - | - |
| | | DA Ensemble (Lee et al., 2020) | 73.2 | - | - |
| | | DA CG (20x, this work) | 81.3 | 66.3 | - |
| | | CIFAR-5m CG (5M, this work) | **83.4** | **71.9** | 86.5 |
| | Finite NN | DA, Small LR, no L2 (Lee et al., 2020) | 83.9 | - | - |
| | | DA, Large LR, L2 (Lee et al., 2020) | 85.6 | | |
| CNN-Pool | Kernel | CNN-LAP Flip (Li et al., 2019) | 82.2 | - | - |
| | | CNN-GAP DA Ensemble (Lee et al., 2020) | 84.8 | - | - |
| | | Myrtle10-Gaussian Flip (Shankar et al., 2020) | 89.8 | 78.3 | - |
| | | Tuned Myrtle10 DA CG (20x, this work) | **91.2** | **79.4** | - |
| | | Myrtle10 CIFAR-5m CG (1.6M, this work) | 89.1 | 79.1 | 89.5 |
| | Finite NN | ResNet18 CIFAR-5m (Nakkiran et al., 2021) | 89.0 | - | 89.4 |
| | | CNN-GAP DA, Small LR, no L2 (Lee et al., 2020) | 84.4 | - | - |
| | | CNN-GAP DA Large LR, L2 (Lee et al., 2020) | 86.7 | | |
| | | Myrtle10 DA (Shankar et al., 2020) | 96.0 | 89.8 | - |

Table 1: CIFAR-10 test accuracy for kernels and finite neural networks of the corresponding architecture type.

et al. (2019); Shankar et al. (2020), a $2\times$ augmentation factor is given by horizontally flipping all images. Beyond that, up to a $20\times$ augmentation factor, we used RandAug (Cubuk et al., 2020) as used in Shankar et al. (2020). In particular for each image in the training set, we sample one ($N = 1$) random augmentation op among ['FlipLR', 'Solarize', 'Color', 'Brightness', 'Contrast', 'Sharpness', 'Posterize', 'Equalize', 'Identity'][6] and apply with magnitude $M = 2$. After applying a random augmentation, we apply a random crop of 4 pixels as well as a random flip. We do not use cutout augmentation. With the strategy, we are able to achieve an accuracy of 91.2% (see Fig. 4). To our knowledge this is the highest published accuracy for a kernel method. Note that while the gap in accuracy between kernels and finite neural networks is reduced, it still remains (see Table 1).

## 5 SEQUENCE AND GRAPH DATA

In this section, we continue to develop neural kernels as a method in their own right. We consider their performance on structured data modalities other than images. Kernels are particularly exciting for such data modalities because we lack inductive biases to take advantage of, and kernel methods such as Gaussian processes are the standard approach for guided experimental design.

First, we consider a protein function prediction benchmark motivated by protein design for targeted gene therapy. There is significant interest in engineering Adeno-associated virus (AAV) capsid proteins to cause the virus to integrate specific DNA into a target cell (Bryant et al., 2021). We perform kernel regression to predict the fitness of mutations across a 28 amino acid window of these proteins, by training and evaluating across all splits, ranging from 82k to 284k examples, of a dataset assaying VP-1 AAV protein variants (Dallago et al., 2021).

Second, we consider a small molecule classification task involving detecting drug cardiac toxicity (Han et al., 2021). This dataset contains descriptions of the graph-structured molecules as SMILES strings along with assays for toxicity. To model this graph structured data, we derive kernels for deep graph neural networks, and fit these to the training set of 6,523 examples.

Developing specific kernels for such data is critical, since traditional covariance functions do not capture task-relevant similarities in high-dimensional inputs with complex structure. The connection between neural networks and neural kernels in the infinite-width limit provides one promising direction to exploit the progress in neural network architecture design. For example, convolutional

---

[6]Specified in https://github.com/modestyachts/neural_kernels_code/blob/master/config.py#L37

| Method | Mut-Des | Des-Mut | 1-vs-rest | 2-vs-rest | 7-vs-rest | low-vs-high |
|---|---|---|---|---|---|---|
| Best reported | 0.79 | 0.75 | 0.48 | 0.74 | 0.74 | 0.39 |
| Ours | 0.81 | 0.84 | 0.64 | 0.71 | 0.71 | 0.34 |

Table 2: Spearman correlation for kernel regression compared to the best result reported for several methods in (Dallago et al., 2021)

| Model | baseline | GP | SNGP | SNGP Ensemble | Ours |
|---|---|---|---|---|---|
| TEST-IID | 0.919 | 0.937 | 0.932 | 0.942 | 0.943 |
| TEST-OOD1 | 0.786 | 0.823 | 0.836 | 0.850 | 0.800 |
| TEST-OOD2 | 0.831 | 0.873 | 0.885 | 0.896 | 0.750 |

Table 3: AUROC for kernel regression compared to other methods reported in (Han et al., 2021).

operations are common in neural networks used on protein sequence data. Similarly, for graph data like molecules, graph convolution operations have grown in popularity. Any such operation has a corresponding kernel that can be used in our approach to kernel regression.

In this vein, for the AAV dataset we considered deep, 1D convolutional architectures and for cardotoxicity we considered deep, graph convolutional architectures. The remaining hyperparamters are depth, weight and bias initialization variance, activation function, diagonal regularization, the type of pooling function, filter size, and whether to use the NNGP kernel or the NTK. To tune hyperparameters for the cardiotox dataset, we used 1k trials on the Google Vizier service (Golovin et al., 2017). Each trial used all examples from the training set with AUROC on the validation set as the objective. For the AAV dataset, tuning hyperparameters using the whole training set is not feasible due to its size, since the kernel must be recomputed each time. Instead we used only 2k examples from the training set with MSE on 1k validation set examples as the objective. Again 1k trials of the Vizier service were used. For additional details on hyperparameters and their tuning, see Appendix C.

After selecting hyperparamters, we ran inference on all splits of the cardiotox and AAV datasets (see Tables. 2 and 3). In all cases, we find our method is either better or competitive with prior methods across all splits.

## 6 CONCLUSIONS

In this work we probe the performance of kernel regression across two dimensions of scale, in the number of examples and the complexity of the kernel. We distributed the computation of large neural kernels with up to 5 million examples and then used distributed conjugate gradients solvers to solve the corresponding linear systems. In doing so, we not only set a new SotA for kernel methods on CIFAR-10 but also outperform highly tuned deep learning methods on a protein design problem and a graph-structured molecule toxicity screening problem. Computational considerations aside, we see that these neural kernels scale remarkably well in terms of predictive performance. Extrapolating from our scaling curves, it appears that significant improvements could still be made by adding more data. These infinitely-wide instantiations of deep neural networks offer some exciting properties and advantages over the parametric, SGD trained alternatives. They offer stable solutions with relatively little optimization and hyperparameter tuning. In the context of Gaussian processes, they represent a distribution over deep networks, with corresponding uncertainty that can be used across decision making tasks such as experimental design. Indeed, our SotA results on protein function prediction for targeted gene therapy and molecule toxicity prediction suggest that these kernels may be very well suited to exciting structured design problems. While computing quantities such as the marginal likelihood or test variances involves additional challenges, one can readily draw samples from the corresponding GP to be used, for example, with Thompson sampling approaches.

The computational cost of computing the neural kernels and solving the corresponding linear systems remains a limitation to scaling further. While there is a significant literature on approximating the kernels with regard to their scaling in the number of examples, approximations of the pairwise kernel computation itself similar to Zandieh et al. (2021); Han et al. (2022) seems necessary.

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

# A  NEURAL KERNEL ARCHITECTURE

Here, we provide details of neural network architectures we use in the experiments.

## A.1  ARCHITECTURE DEFINITION

For neural network architecture for our kernel used in the experiments, we provide layer specification code using Neural Tangent (Novak et al., 2020) library.

**Fully-Connected Network (FC):**  We use vanilla 3-hidden fully-connected layers with ReLU activation function. We denote this neural architecture as `FC3`.

```python
from neural_tangents import stax

def FC3():
  depth = 3
  W_std = 2.0**0.5
  b_std = 0.1
  num_classes = 10
  dense = functools.partial(stax.Dense, W_std=W_std, b_std=b_std)
  layers = [stax.Flatten()]
  for _ in range(depth):
      layers += [dense(512), stax.Relu()]
  layers += [dense(num_classes)]

  return stax.serial(*layers)
```

**Convolutional Neural Network with Vectorization aggregation (CNN-VEC):**  We us vanilla 8-hidden convolutional layers with ReLU activation function. After the convolutional layers, aggregation is done by flattening (or vectorization). We denote this neural architecture as `CV8`.

```python
from neural_tangents import stax

def CV8():
  depth = 8
  W_std = 2.0**0.5
  b_std = 0.1
  num_classes = 10
  conv = functools.partial(stax.Conv, W_std=W_std, b_std=b_std,
                           padding='SAME')
  layers = []
  for _ in range(depth):
      layers += [conv(512, (3, 3)), stax.Relu()]
  layers += [stax.Flatten(),
             stax.Dense(num_classes, W_std=W_std, b_std=b_std)]

  return stax.serial(*layers)
```

**Myrtle Convolutional Neural Network (Myrtle):**  The neural kernel for Myrtle CNN was first introduced by Shankar et al. (2020). Using intermediate average pooling layers, the convolutional architecture is more efficient than global average pooling while achieving high performance with 10-layers.

```python
from neural_tangents import stax

def Myrtle():
  W_std = 2.0**0.5
  b_std = 0.1
  num_classes = 10
```

```python
    activation_fn = stax.Relu()
    conv = functools.partial(stax.Conv, W_std=W_std, b_std=b_std,
                             padding='SAME')
    layers = [conv(512, (3, 3)), activation_fn]
    layers += [
        conv(512, (3, 3)), activation_fn] * 2
    layers += [stax.AvgPool((2, 2), strides=(2, 2))]
    layers += [
        conv(512, (3, 3)), activation_fn] * 3
    layers += [stax.AvgPool((2, 2), strides=(2, 2))]
    layers += [
        conv(512, (3, 3)), activation_fn] * 3
    layers += [stax.GlobalAvgPool()]
    layers += [
        stax.Flatten(),
        stax.Dense(
            num_classes,
            W_std,
            b_std)
    ]

    return stax.serial(*layers)
```

**Tuned Myrtle Convolutional Neural Network (Tuned Myrtle)**

```python
from neural_tangents import stax

def TunedMyrtle():
  W_std = 2.0
  b_std = 0.01
  num_classes = 10
  activation_fn = stax.ExpNormalized()
  norm_layer = stax.LayerNorm(axis=(1, 2, 3))
  conv = functools.partial(stax.Conv, W_std=W_std, b_std=b_std,
                           padding='SAME')
  layers = [conv(512, (2, 2)), norm_layer, activation_fn]
  layers += [conv(512, (3, 3)), norm_layer, activation_fn] * 2
  layers += [stax.AvgPool((2, 2), strides=(2, 2))]
  layers += [conv(512, (3, 3)), norm_layer, activation_fn] * 4
  layers += [stax.AvgPool((2, 2), strides=(2, 2))]
  layers += [conv(512, (3, 3)), norm_layer, activation_fn] * 2
  layers += [stax.GlobalAvgPool(), stax.Flatten()]
  for _ in range(2):
    layers += [stax.Dense(512, W_std, b_std),
               stax.LayerNorm(), activation_fn]
  layers += [
      stax.Dense(
          num_classes,
          W_std,
          b_std)
  ]
  return stax.serial(*layers)
```

**Simple Graph Neural Network**  Our simple graph neural network is based off Graph NTK (Du et al., 2019) where the aggregation node combines node features according to the graph connectivity.

```python
from neural_tangents import stax

def SimpleGNN(config):
```

```
    depth = 25
    W_std = 0.6
    b_std = 0.1
    W_std_out = 0.1
    b_std_out = 0.03
    num_classes = 2
    activation_fn = stax.Erf()
    dense = functools.partial(
        stax.Dense, W_std=W_std, b_std=b_std)
    aggregate = stax.Aggregate(aggregate_axis=1, batch_axis=0, channel_axis=2,
                               implementation='DENSE')
    layers = []
    for _ in range(depth):
      layers += [dense(512), activation_fn, aggregate]
    layers.append(stax.GlobalAvgPool())
    layers.append(stax.Dense(num_classes,
                             W_std=W_std_out, b_std=b_std_out))
    return stax.serial(*layers)
```

**1D Convolution Neural Network**

```
from neural_tangents import stax

def OneDimConv(config):
  depth = 5
  W_std = 1.7
  b_std = 0.5
  W_std_out = 1.0
  b_std_out = 0.
  dimension_numbers = None

  filter_size = 15
  filter_tuple = (filter_size,)
  use_gap = True

  padding = 'SAME'
  parameterization = 'ntk'
  activation_fn = 'erf'

  collect_layers = []
  conv_or_lcn = functools.partial(
      stax.Conv, W_std=W_std, b_std=b_std, padding=padding,
      parameterization=parameterization)
  if depth > 0:
    collect_layers += [
        conv_or_lcn(512, filter_tuple, dimension_numbers=dimension_numbers),
        activation_fn
    ]
  for _ in range(1, depth):
    collect_layers += [conv_or_lcn(width, filter_tuple), activation_fn]

  collect_layers += [stax.GlobalAvgPool()]

  collect_layers += [
      stax.Flatten(),
      stax.Dense(1, W_std_out, b_std_out, parameterization=parameterization)
  ]

  return stax.serial(*collect_layers)
```

## A.2 Regularized ZCA preprocessing

For convolutional kernels on image data, (e.g. `CV8`, `Myrtle`, `Tuned Myrtle`), we preprocess inputs with regularized ZCA whitening. Following the convention of Lee et al. (2020), for flattened $d$-dimensional, $n$ training inputs $X \in \mathbb{R}^{d \times n}$, the input data covariance is $\Sigma_X = \frac{1}{d} X X^T$. Considering the SVD of $\Sigma_X = U D U^T$, the regularized ZCA whitening transform is given by

$$W_{\text{ZCA}} = U \sqrt{\left( D + \epsilon \frac{tr(D)}{d} I_d \right)^{-1}} U^T, \qquad (1)$$

where $\epsilon$ is the regularization parameter.

It has been demonstrated that regularzied ZCA significantly helps performance of convolutional neural kernels on image classification tasks (Shankar et al., 2020; Lee et al., 2020; Nguyen et al., 2021b). For `CV8` we use $\epsilon = 3.0$, and for `Myrtle` and `Tuned Myrtle` we use $\epsilon = 0.1$.

## B  Sequence data preprocessing

The AAV data consists of sequences of amino acids of different lengths (around 700). The sequences were one-hot encoded and padded with a special character so they were all the same length as the longest sequence. Finally the features and labels were normalized to have mean 0 and standard deviation 1.

## C  Architecture hyperparameter tuning

### C.1  Tuning graph convolution architecture on Cardiotox

The activation function was tuned among ReLU, Erf, Identity, GeLU, Normalized Exponential, and Sigmoid. The kernel type was tuned among the NNGP kernel and the NTK. The weight (except the final layer) initialization standard deviation was tuned in $[0.1, 2.5]$ on a linear scale. The bias initialization standard deviation was tuned among $\{0., 0.001, 0.003, 0.01, 0.03, 0.1, 0.3\}$. The last-layer weight initialization standard deviation was tuned in $[0.1, 10.0]$ on a logarithmic scale. The depth was tuned among $\{1, 2, 3, 5, 7, 9, 10, 12, 15, 20, 25, 30, 50\}$. Whether to include self-loops was also tuned. The diagonal regularization term was tuned among

```
import numpy as np
np.concatenate([[0.], np.logspace(-10, 4, 49)])
```

### C.2  Tuning 1D convolution architecture on AAV

The activation function was tuned among ReLU, Erf, Identity, GeLu, and Normalized Exponential. The kernel type was tuned among the NNGP kernel and the NTK. The weight (except the final layer) initialization standard deviation was tuned in $[0.1, 2.5]$ on a linear scale. The bias initialization standard deviation was tuned among $\{0., 0.01, 0.05, 0.1, 0.5\}$. The last-layer weight initialization standard deviation was tuned in $[0.1, 10.0]$ on a logarithmic scale. The depth was tuned among $\{1, 2, 3, 5, 7, 9\}$. The diagonal regularization term was tuned among

```
import numpy as np
np.concatenate([[0.], np.logspace(-10, 4, 49)])
```

As mentioned in the main text, only a small subset of the training data was used for tuning. To investigate the sensitivity of hyperparameters to dataset size, we ran inference for several small dataset sizes and check the correlation of the objective (see Fig. 5). We saw good correlation (0.918) and rank agreement (Spearman's rank 0.700) for test MSE when 100 vs 2k training examples were used. This suggests the choosing hyperparamters based on a smaller subset of the training set is reasonable.

### C.3  Tuning Myrtle architecture on CIFAR-10

The activation function was tuned among ReLU, Erf, Identity, GeLU, Normalized Exponential and its variants. Whether to include layer normalization after affine layer and whether to use

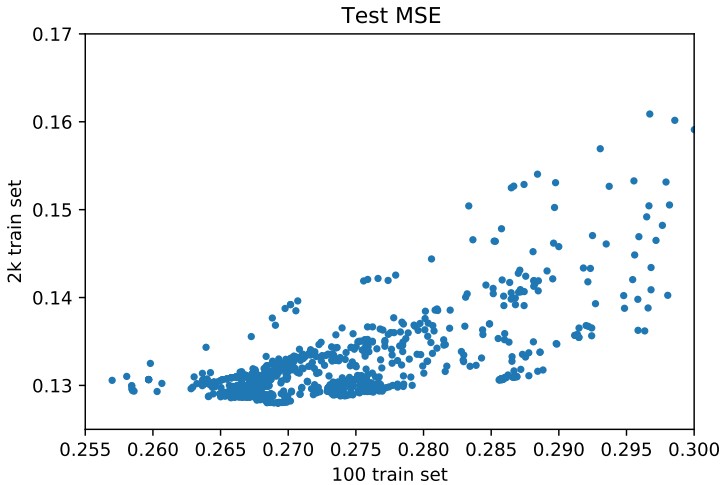

Figure 5: A scatter plot of test MSE. Each point is a different assignment of the hyperparameters whose coordinates are the test MSE using 100 and 2k examples to train respectively.

global average pooling after the convolution layer was tuned. The weight initialization standard deviation was tuned in $[0.1, 2.5]$ on a linear scale. The bias initialization standard deviation was tuned among $\{0., 0.001, 0.003, 0.01, 0.03, 0.1, 0.3\}$. The first convolution's filter size was tuned among $\{1, 2, 3, 4, 5, 8, 10\}$ and the remaining convolutional layers' filter size was tuned among $\{1, 2, 3, 4, 5, 8\}$. The number of convolutional layers before each $(2, 2)$ average pooling layer was tuned among $\{0, 1, 2, 3, 4, 5\}$. After convolutional layers, option to add $\{0, 1, 2\}$ dense layers was tuned. ZCA regularization strength was tuned in $[0.003, 10.0]$ on a logarithmic scale. As described in the tuning of the AAV dataset, a small subset of 1,280 training examples were used for tuning.

Among these, the most robust and high-performing setting was selected for the Tuned Myrtle architecture used for CIFAR-10 data augmentation and Tiny ImageNet experiments.

