# OpenReview forum: "Kernel Regression with Infinite-Width Neural Networks on Millions of Examples"
_ICLR.cc/2023/Conference — Submitted to ICLR 2023_

### Official Review · Reviewer_7cjF · 2022-10-23

**Confidence:** 4
**Correctness:** 4
**Technical Novelty And Significance:** 3
**Empirical Novelty And Significance:** 3
**Recommendation:** 8

**Clarity, Quality, Novelty And Reproducibility:**

The paper is very well written and the challenges to move from the Gaussian process setting to the neural tangent kernel are clearly described. The experiments are well-motivated and very relevant to the community. What remained unclear to me is how many GPUs/CPUs, RAM etc are needed for certain types of experiments. How many GPUs does one need to train on CIFAR10 with augmentations for instance? Including these numbers would be helpful for the reader who plans to use the framework for his own experiments.
As mentioned above, this work is largely built on top of the work [1], which reduces the originality of the framework. The obtained extension still remains non-trivial however.

[1] Ke Wang, Geoff Pleiss, Jacob Gardner, Stephen Tyree, Kilian Q Weinberger, and Andrew Gordon Wilson. Exact gaussian processes on a million data points. In Advances in Neural Information Processing Systems, 2019.

**Strength And Weaknesses:**

**Strengths**:
1. The paper is very well-written and very easy to follow. I really enjoyed reading this paper.
2. The resulting methodology in my opinion enables a lot of interesting experiments (provided enough compute) that were simply infeasible before. The obtained results provide a better and more fair comparison between finite and infinite networks, showing for instance that when standard training components such as data augmentation are included for kernels, very competitive accuracies can be achieved with the kernel method as well. It also marks a first step towards evaluating such kernels on ImageNet, which further facilitates comparisons.
3. The experiments regarding scaling laws are very interesting and important as they provide a more fine-grained look into performance. To my knowledge, such experiments are missing in the literature. It would have been even more interesting to compare the kernel scaling laws with the corresponding finite network.

**Weaknesses**
1. The methodology largely relies on previous work (i.e. [1]) with the only difference that the kernel is stored on disk instead of computed on the fly. While the smart reading strategy of the chunks is certainly not trivial, the technical novelty of the approach is somewhat limited.
2. While the protein/molecule experiments are very interesting in their own right, I don’t see how they fit into this paper. Most datasets are only moderately large and the involved methodology developed in this paper is thus not needed to perform accurate inference.

[1] Ke Wang, Geoff Pleiss, Jacob Gardner, Stephen Tyree, Kilian Q Weinberger, and Andrew Gordon Wilson. Exact gaussian processes on a million data points. In Advances in Neural Information Processing Systems, 2019.

**Summary Of The Paper:**

This work enables kernel regression with the so-called neural tangent kernel to work with significantly larger datasets compared to before (order of million datapoints) by building on top of the framework of [1] for Gaussian processes. The key difference to [1] stems from the difference of computational costs to obtain the kernel matrix; while for GPs, simpler kernels such as the RBF are often used, calculating the Gram matrix of the NTK associated with a convolutional neural network marks the bottleneck, instead of the linear system that needs to be solved. The authors distribute the computation of the kernel over many devices and store it on disk. To perform the matrix-vector product needed in conjugate gradient descent, the authors use a smart way to read chunks of the kernel efficiently, avoiding thus the need to recompute the kernel at every step. The more powerful method enables experiments that were not possible before, such as large scaling law experiments, experiments on TinyImageNet and heavy data augmentation leading to SOTA kernel performance on CIFAR10. Finally, the authors also show that these kernels can reach very strong performance on smaller datasets including molecular tasks.

**Summary Of The Review:**

In summary, I would like to see this work accepted at ICLR. The framework developed in this work enables important experiments with NTK that were infeasible before. Scaling laws, usage of data augmentation and the possibility to scale to larger datasets such as TinyImageNet facilitate comparing finite and infinite width networks, which still remains an open problem.

---

> ### Author Response · Authors · 2022-11-15
> **Response to Reviewer 7cjF**
>
> We are grateful for Reviewer 7cjF’s time and effort and are glad they would like to see our paper accepted. We have improved our paper based on the reviewer's comments.
>
> **[Methodological contribution]:** Please see our comments above to all reviewers.
>
> **[Protein/molecule experiments]:** Our aim here was to move away from the high-SNR image datasets to which finite CNNs are especially suited to investigate whether neural kernels might outperform. The reviewer is correct about the CardioTox dataset. However, the AAV benchmark would be very challenging without the additional parallelization of the kernel computation we implemented. The largest dataset previously considered for this type of kernel was 100K, whereas AAV has 284K data points. To address this concern about the small CardioTox dataset, we considered instead the MOLPCBA dataset, which has 438K data points. Again we find that neural kernels are very competitive. More details are included above.
>
> **[Wang et al.]:** Please see our comments above to all reviewers.

---

### Official Review · Reviewer_wb1p · 2022-10-25

**Confidence:** 4
**Correctness:** 3
**Technical Novelty And Significance:** 1
**Empirical Novelty And Significance:** 2
**Recommendation:** 3

**Clarity, Quality, Novelty And Reproducibility:**

This paper is very clear in what approach it takes and what the result is. However, it is unclear what the central message is. The originality is limited given the prior work on scaling GPs (the authors might want to expand the discussion on distributed computation to show if there is originality there).

**Strength And Weaknesses:**

## Strengths

* The paper makes a nice attempt to scaling infinite-width neural-network kernels. This is an important problem faced by the area and a solution would be of interest to a wide audience.

* The empirical study is comprehensive and covers several data modalities including images, sequences and graphs, all showing competitive performance against strong baselines.

## Weaknesses

* The results are good, but not surprising. Given the existing work of Wang et al. (2019) that already applies GPs to 1 million data points, the scaling in this work was not that far from reach. In fact, the paper is using the same preconditioning as Wang et al. (2019) and from the text I did not see any additional innovations needed to achieve the 5 million scale. Achieving 91% on CIFAR 10 with data augmentation that increases the training set by a factor of 20 does not seem surprising, either.

* One big question i had in mind while reading this paper is---what is the central message? The focus in each section seems completely orthogonal to each other and the whole piece of work feels more like an experiment report. Simply showing that infinite-neural-network kernels achieves predictable good (but not state-of-the-art) results does not qualify a scientific contribution.

* Some details are unclear about the 5 million experiment. What are the tolerance level needed to solve this linear system? One big concern I had about Wang et al. (2019) is that they used a very large tolerance (1 if I remember it correctly) during training and did not bother to get the GP hyperparameters right (partly the reason why the training is super fast/scalable). During test time, to ensure good performance, they had to switch to a much smaller tolerance level. Do you have to do the same? What is total training + test time when compared to a standard sparse method like Nystrom? (Nystrom-like ideas  is not totally infeasible for these kernels, see Deng, Z., Shi, J., & Zhu, J. (2022). NeuralEF: Deconstructing Kernels by Deep Neural Networks. arXiv preprint arXiv:2205.00165
where they show a nice recovery of NTK using sparse eigenfunctions.


**Summary Of The Paper:**

This paper empirically investigates the performance of million-scale kernel regression with infinite-width neural networks using the preconditioned conjugate gradient method from the Gaussian process literature (Wang, 2019). The method is applied to several domains including testing scaling laws on CIFAR-5m, enabling data augmentation on CIFAR-10 by a factor of 20, and sequence/molecule classification tasks using convolutional/graph convolutional architectures.

**Summary Of The Review:**

Given the lack of central message and limited amount of contribution compared to prior literature, I would recommend rejection.

---

> ### Author Response · Authors · 2022-11-15
> **Response to Reviewer wb1p**
>
> We thank Reviewer wb1p for taking the time to read our paper and suggesting ways it might be improved. We are slightly confused by Reviewer wb1p’s score given that they feel the paper addresses an “important problem [...] of interest to a wide audience” and is a “comprehensive study showing competitive performance against strong baselines.” Hopefully we address many of the concerns below and hope the score can be raised.
>
> We address many of this reviewer’s concerns above in our comment to all reviewers.
>
> **[Results are not surprising]:** While the results may not be surprising to this reviewer, the goal of the data augmentation experiments was to understand how much the gap in performance could be closed. Regardless of the result, this was not known before our work and adds an important result to the literature. The fact that we achieve 91% accuracy on CIFAR-10, which is SotA for a kernel method, is an added bonus. We believe that our results show the potential ceiling that is achievable with the addition of data augmentation to neural kernels, and show that it is not possible to naively close the performance gap with finite neural networks with this approach alone.
>
> **[Not SotA]:** Please note our results on AAV benchmark are SotA when compared to all other methods, not just kernels. We do not feel this result was at all predictable before our work.
>
> **[Tolerance]:** First, we note that for kernel regression (as opposed to GPs) there is no notion of inference tolerance. We simply solve the linear system to a specific tolerance. In most of our experiments, we used a tolerance of 1e-4. However, we found that most metrics evaluating the predictions on the testset, such as accuracy, stabilized well before the residual was approximately 1e0, as in Wang et al. Pushing the residual below 1e-4 from 1e0 takes many additional steps of CG. This was not feasible for the 5 million experiments. We will add additional details on tolerance to the appendix.
>
> **[Nystrom]:** We agree that Nystrom approximation is not infeasible, and indeed we consider it in Section 2. Since our goal is to investigate the maximum potential of neural kernels, we do not compare their running times, only their performance. We hope that future work will produce better ways of approximating neural kernels. Note that our approach is not in competition with but is complementary to low-rank approximations. Nystrom approximations still require solving a large linear system when rank increases, which can be achieved with preconditioned CG. We feel that our results highlight an important message to the community: we need efficient approximations to the expensive pairwise neural kernel-functions as well as efficient approximations to the NxN kernel.

---

### Official Review · Reviewer_c8Ru · 2022-11-01

**Confidence:** 4
**Correctness:** 2
**Technical Novelty And Significance:** 1
**Empirical Novelty And Significance:** 2
**Recommendation:** 5

**Clarity, Quality, Novelty And Reproducibility:**

- **Clarity**: The paper reads well.
- **Quality**: Experimental results could be strengthened by adding error bars and running durations.
- **Novelty**: The main idea here, namely to parallelize the computation of the Gram matrix and store it to disk, isn't particularly novel.
- **Reproducibility**: While the authors provided Python code, specifications of hardware used would have eased reproducibility.


**Strength And Weaknesses:**

- **Strength**: Scalability is one of the major hurdles in bridging the gap between  GP methods and deep learning. This paper's attempt to doing so is laudable, and experimental results are encouraging.

- **Weaknesses**: Experimental results seem to be missing error bars/intervals, hardware specs, and running durations. Overall the contribution seems too light to be worthy of publication.




**Summary Of The Paper:**

This paper explores further scaling up preconditioned conjugate gradient (PC-CG) based GP regression by parallelizing the computation of the Gram matrix as a one-off and storing it to disk in "sub-blocks" to be loaded in the RAM in a multi-threaded fashion as/when needed.

This contrasts with standard PC-CG where the Gram matrix is never computed entirely but rather partially recomputed on the fly as needed.

The standard PC-CG approach works reasonably well when the kernel is easy to compute (e.g. the square-exponential kernel), but breaks down when the kernel is more complex.

The authors demonstrate the effectiveness of the proposed approach using neural kernels, especially compared to alternatives consisting of approximating the linear system (as opposed to solving it within an error CG-style), and propose additional empirical studies such as how performance of GP regression under neural kernels scale with data sizes.


**Summary Of The Review:**

Overall the main contribution is too simple to be worthy of a paper and, in any case, empirical evaluation should be strengthened.

---

> ### Author Response · Authors · 2022-11-15
> **Response to Reviewer c8Ru**
>
> We are grateful to Reviewer `c8Ru` for their effort reviewing our paper. We are glad Reviewer `c8Ru` agrees that our results are encouraging and appreciates our efforts tackling the main obstacle of scalability for kernel methods. Below we emphasize the main contributions of our work and ask Reviewer `c8Ru` to reconsider their score in light of this.
>
> **[Error bars]:** Note that our method is not randomized given the sample of training data. Normally the error bars reported for DL methods are with respect to the random seed, which we do not have here. Can the reviewer elaborate on what kind of error bars would be useful?  One possibility is by resamping the training data a variance can be estimated. This can work for small dataset sizes but naturally cannot be used near the full dataset size. Another possibility is to plot predictive uncertainty, which we do not have at the moment since we are only solving kernel regression (instead of full GP regression). While there are ideas to obtain predictive uncertainty as described in our conclusion, it is out of scope and hopefully an interesting future project.
>
> **[Hardware and running times]:** Please see our general comments on methodological contributions to all reviewers. Our goal was to investigate the potential of neural kernels as opposed to optimizing running time and hardware usage.

---

### Official Review · Reviewer_nR21 · 2022-11-03

**Confidence:** 4
**Correctness:** 4
**Technical Novelty And Significance:** 3
**Empirical Novelty And Significance:** 4
**Recommendation:** 6

**Clarity, Quality, Novelty And Reproducibility:**

The paper is well-written, with clear presentation of experimental results and
ample references to prior work. The work enables new applications of neural
kernels at scale (Tiny ImageNet) and in various applications (the performance
gains over traditional kernels give justification for the authors' work in
these contexts).

It would be preferable for reproducibility to have code released to reproduce
experiments, rather than just the architecture specifications used to generate
the kernels (appendix A).



**Strength And Weaknesses:**

## Strengths

- The engineering contribution directly enables new evaluation of neural
  kernels at novel scales, enabling their use on new datasets where they set
  new record for performance by kernel methods.
- The paper is well-written and does a good job of organizing various
  approaches to large-scale kernel regression in section 1.2 and section 2. One
  comes away with an excellent understanding of the landscape of approaches to
  this problem, which should be useful for future work.
- The authors do a good job of showing how the engineering advancements open
  new opportunities for kernel methods. They demonstrate that neural kernels
  achieve excellent performance on molecular prediction/classification tasks
  compared to existing methods, and show that data augmentation can give good
  improvements in performance for kernel methods, given the ability to work
  with larger kernels.

## Weaknesses

- The precise methodological contributions seemed somewhat unclear to me -- is
  the main technical contribution to identify the most relevant existing
  large-scale kernel regression approaches for working with neural kernels (as
  outlined in section 2), and then to develop and implement a new parallelization
  scheme on top of these methods to facilitate computing neural kernels? The
  latter feels like a somewhat limited (but nonetheless significant) technical
  contribution, given that the paper's description of the parallelization scheme
  seems to be mostly restricted to showing Figure 1.
- It might be better to have a demonstration of the method on low-resolution
  datasets other than CIFAR-5m, since this dataset is essentially synthetic. I
  think the authors' experiments on Tiny ImageNet are more than appropriate in
  this connection, I just point this out because the description of CIFAR-5m at
  the bottom of page 5 as "...ensuring that the additional data are sampled
  i.i.d." is not correct and somewhat misleading (since the dataset is
  generated from a trained generative model).
- I think the section about neural scaling laws could be improved
  by a more precise discussion of the issues -- once one is working with linear
  models (whether or not the kernel is "neural"), one can make much more
  precise statements about the asymptotic scaling of the test error as a
  function of the number of samples given various distributional assumptions,
  and it does not seem to be appropriate to me to treat this as a purely
  empirical endeavor.
  These types of questions have been studied in the literature on nonparametric
  regression in RKHSes; under simple distributional models for the data (e.g.,
  the target function is sufficiently smooth and the data come from a
  distribution on a $d$-dimensional manifold), the RKHS associated to a given
  kernel can be characterized in terms of differentiability properties of the
  kernel (which are, in turn, inherited from the architectural choices and the
  random initialization scheme), and the corresponding scaling behavior (i.e.,
  the exponent) of the test error becomes a function of this RKHS, the
  dimension $d$, and the target function's smoothness.


**Summary Of The Paper:**

The authors consider the large-scale implementation of 'neural kernels', kernel
methods that are derived from the architecture of a specific neural network and
the correspondence between randomly-initialized neural networks and gaussian
processes. In contrast to existing large-scale kernel methods for use with
kernels like the RBF kernel, the fact that these neural kernels are
computationally-expensive to compute necessitates additional parallelization.
The authors review methods for large-scale kernel regression, implement their
own method in the context of neural kernels, and show experiments on various
large datasets (CIFAR-5m, Tiny ImageNet) where these kernels achieve
performance beyond past kernel methods. They also demonstrate their approach on
various basic science tasks, where it outperforms existing kernel methods.



**Summary Of The Review:**


The engineering contribution and its consequences are significant, and the
authors verify these well through various experiments. One hopes the code will
be released.

---

> ### Author Response · Authors · 2022-11-15
> **Response to Reviewer nR21**
>
> We are grateful to Reviewer nR21 for taking the time to read our manuscript and provide feedback. We are pleased that the reviewer agrees the paper contains significant engineering contributions and consequential results. We will try to answer some specific questions raised and hope the reviewer will consider increasing their score.
>
> **[Methodological contributions]:** Please see our general comments on this to all reviewers.
>
> **[Additional image datasets]:** Unfortunately, we are not aware of other standard benchmark datasets at low-resolution with as many samples as CIFAR-5m. Moreover, CIFAR-5m was specifically generated for this purpose of studying models as more training data are added. We also want to emphasize that the data are i.i.d., since each sample from a generative model is i.i.d. for a different seed. This statement is of course conditional on the pre-trained generative model. Note that we also test on data from CIFAR-5m. This treats CIFAR-5m as a dataset in its own right, irrespective of its relationship to CIFAR-10.
>
> **[Theoretical results on scaling laws for kernels]:** We agree with the reviewer that this question is not entirely empirical. However, the results we are aware of make strong assumptions on the kernel or data distribution that often cannot be verified in practice. While the question is not purely empirical, we feel this does not undermine the utility of experimental work.
>
> **[Code]:** We would like to open source as much of the code as possible. Indeed many of the libraries we use are already open source, e.g. neural_tangents, JAX, and Courier. However, much of the code is highly specific to our infrastructure. Moreover, we feel much of the code will be of limited use due to the sheer size of the computation and data storage requirements (e.g. 200TB kernel matrix).

---

> > ### Comment · Reviewer_nR21 · 2022-12-06
> > **thanks**
> >
> > Dear authors,
> >
> > Thanks for your response to my review. Here are some specific acknowledgments of your response:
> > - [Datasets] I see your point about CIFAR-5m being a dataset in its own right once we have 'conditioned' on the specific generative model, and agree with you after this specific qualification. I just found the discussion of this issue in your submission (second to last graf at the bottom of page 5) to be somewhat less clear about this issue -- I read the statement "...while ensuring the additional data are sampled i.i.d." as implying a connection to the original distribution of CIFAR-10, since it was mentioned at the start of the graf, although evidently you are not confused about this issue yourselves. Perhaps some additional words of clarification in the submission would make this completely clear to the reader.
> > - [Scaling laws] I agree with your point of view here, but I still feel that the issue would be presented more circumspectly in the submission if  some specific theoretical works were mentioned here. This would also seem to serve to make your point about the use for empirical studies here more convincingly, in my eyes.
> > - [Code] On this point I am slightly less convinced, but I may be failing to appreciate some standards for research in this area. I can appreciate that your specific tooling will make reproducing the experiments a challenge, even with the code in hand, but I feel the submission should present your methodology clearly enough that it is possible in principle for an interested and motivated reader (with a sufficient hardware budget...) to re-implement your framework, and it does not seem to me that the presentation in the current submission is sufficient here.

---

> > > ### Author Response · Authors · 2022-12-08
> > > **Discussion continued**
> > >
> > > Thank you again for your comments! We agree that clarifying the discussion on the CIFAR-5m dataset and adding references to theoretical work when discussing scaling laws will improve the paper. We will update the manuscript to reflect this.
> > >
> > > We also agree that it is important for readers to be able, in principle, to reimplement and reproduce our results. This would require both computing the kernel and then performing inference. For computing the kernel, the code shared in the appendix contains the crucial information. However, we will include additional details about how we scale and distribute this computation for larger kernels—of course, the particular implementation here is very infrastructure dependent. For inference, we believe we can open source the core code of preconditioned conjugate gradients. Again, the size of problem this can be used on will be limited by the hardware that is available. Finally, we would also like to share the kernels that we have already computed with the research community. We are looking into ways to do this, but have not yet found a practical solution due to the massive storage requirements. However, if sharing the whole kernels proves impossible, sharing those for smaller datasets or just a subset of the largest kernels is an option.

---

### Author Response · Authors · 2022-11-15
**Response to all reviewers**

**Main message of the paper**

The tremendous revolution we're seeing in deep learning has been largely due to training and modeling at scale. Up to this point, no one knew if kernels would yield similar results if pushed to the limits. We are pushing the limits along two axes, the number of examples and the expressivity of the kernels, and find that we can compete with DL but both are needed.

Our aim was to investigate neural kernels in two directions. For the first direction, because of their connection to finite-width neural networks, neural kernels can be used to better understand deep learning. The most significant difference between finite neural networks and neural kernels is that the latter lack any feature learning. Feature learning remains the least theoretically understood part of deep learning. An important question is, how much is the gap in performance between these two model classes due to feature learning? We study this in two settings: 1) Do these methods perform differently as more data are added, and is there a difference in their scaling exponents? This question is especially important in the large model era. While scaling exponents are better theoretically understood for kernels, often these results rely on assumptions that either do not hold for complex neural kernels or cannot be verified in practice—emphasizing the need for empirical work. 2) How much of the performance gap on standard benchmarks can be closed by data augmentation? While we agree that it is not surprising that performance of neural kernels improves with data augmentation, it is still significant to establish by how much. Previous kernel-based approaches have achieved significant improvements in performance by augmenting the training dataset, and it is important to study whether enough data augmentation could successfully bridge the gap between neural kernels and finite neural networks, or if there are smarter inductive biases and feature learning that must be incorporated.

The second direction we consider is whether neural kernels have potential as a method in their own right. Moving away from the high-SNR image classification datasets to which CNNs are especially suited, we find they have significant potential. On the large-scale protein function prediction dataset, AAV, we find that they either outperform or are competitive with all methods (not just kernels). To emphasize this point, we have run additional experiments on the MOLPCBA dataset from the Open Graph Benchmark. This dataset contains approximately 400K molecules that have been assayed in a variety of ways. Again, we find that our method is competitive with SotA methods. The SotA without additional data is an mean average precision of 0.3012 (https://ogb.stanford.edu/docs/leader_graphprop/#ogbg-molpcba), and our simple GNN kernel obtains 0.2651. Another point of comparison is 0.2020 obtained by a finite-width GNN without additional tricks. We will update the manuscript with more details.

While there are many more possible innovations to make neural kernels more practical at scale, we feel our results are the first step in this direction and demonstrate they have clear potential as a stand-alone method. Perhaps most promising is that with some additional work, they can be used as the kernel in a Gaussian process to produce a fully-Bayesian method.

---

> ### Author Response · Authors · 2022-11-15
> **Response continued**
>
> **Methodological contribution**
>
> We emphasize that we are not claiming novel methodological contributions and do not feel this is necessary for a significant contribution to ICLR. We feel the results contribute new, valuable knowledge to the community. This includes information about the importance of feature learning and data augmentation for neural networks and on scaling laws, which as Reviewer `7cjF` points out is missing from the current literature.
>
> In addition to these contributions on understanding deep learning, we investigate the potential of neural kernels. We acknowledge that many more innovations are needed for neural kernels to compete with neural networks on practicality. While we did investigate low-rank approximations to the linear system, we leave these innovations to future work. Other than making the experiments feasible for us to complete, we did not attempt to optimize running times and so do not report them. As stated in the manuscript, the main computational cost is in computing the kernel, which scales quadratically with dataset size.
>
> The primary challenge is in applying previous methods at a much larger scale. The complex neural kernels, such as myrtle10, had only been used on 1e5 data points previously. That is, 5e9 kernel entries and ~40GB of data (symmetric matrix at f64). Applying the same kernel to 5e6 data points implies ~1.3e13 entries and ~100TB of data. This many entries exposes the routine to very rare errors. Even single incorrect entries can mean the kernel matrix is no longer positive definite, causing CG to fail. Parallel IO for this volume of data is also nontrivial. As stated by Reviewer `nR21`, solving such problems is a significant technical contribution.
>
> In addition, we apply neural kernels to a protein prediction and two small molecule benchmarks and demonstrate their potential in these areas. Again, our results on many splits of the AAV benchmark are state-of-the-art when compared to all other methods, not just kernels.

---

> > ### Author Response · Authors · 2022-11-15
> > **Response continued**
> >
> > **Comparison on Wang et al., 2019**
> >
> > Many reviewers mentioned [Wang et al., 2019] in relation to our work, which we cite and acknowledge. However, there are several differences we want to highlight. The most significant for our work is that Wang et al. focuses on simpler kernels that are cheap to compute, stationary, and lead to better conditioned linear systems, in contrast to neural kernels. Moreover, Wang et al. has additional contributions in computing the full GP posterior, not only its mean. For example, they develop an estimator for the marginal likelihood. Indeed the preconditioned conjugate gradients algorithm is well known and has been used to solve large linear systems for many years (Sec. 8.3.6, Rasmussen and Williams, 2006). We faced additional challenges in our work posed by the computational expense of neural kernels.
> >
> > A further difference is that our largest dataset contains 5 million points and theirs 1.3 million points. This implies a kernel that is approximately 15x larger—a completely nontrivial difference, and requires 56x more compute to solve with naive cubic scaling.

---

### Decision · Program_Chairs · 2023-01-20

**Decision:**

Reject

**Justification For Why Not Higher Score:**

The major concern, shared by all reviewers, is that this paper does not have enough technical details to reproduce the experimental results. The AC agrees and thus recommends rejection.

**Justification For Why Not Lower Score:**

N/A

**Metareview: Summary, Strengths And Weaknesses:**

This paper combines a series of techniques to improve kernel methods on large-scale datasets and achieves impressive results.
The major concern, shared by all reviewers, is that this paper does not have enough technical details to reproduce the experimental results. The AC agrees and thus recommends rejection.